# Mobile phone-based interventions for mental health: A systematic meta-review of 14 meta-analyses of randomized controlled trials

Simon B. Goldberg[1,2]*, Sin U Lam[1,2], Otto Simonsson[2], John Torous[3], Shufang Sun[4,5]

1 Department of Counseling Psychology, University of Wisconsin-Madison, Madison, WI, United States of America, 2 Center for Healthy Minds, University of Wisconsin-Madison, Madison, WI, United States of America, 3 Division of Digital Psychiatry, Beth Israel Deaconess Medical Center, Harvard Medical School, Boston, MA, United States of America, 4 Department of Behavioral and Social Sciences, Brown University School of Public Health, Providence, RI, United States of America, 5 Mindfulness Center, Brown University, Providence, RI, United States of America

* sbgoldberg@wisc.edu

**Data Availability Statement:** Study-level data are reported in included studies for this review (see tables and citations for all included studies).

## Abstract

Mobile phone-based interventions have been proposed as a means for reducing the burden of disease associated with mental illness. While numerous randomized controlled trials and meta-analyses have investigated this possibility, evidence remains unclear. We conducted a systematic meta-review of meta-analyses examining mobile phone-based interventions tested in randomized controlled trials. We synthesized results from 14 meta-analyses representing 145 randomized controlled trials and 47,940 participants. We identified 34 effect sizes representing unique pairings of participants, intervention, comparisons, and outcome (PICO) and graded the strength of the evidence as using umbrella review methodology. We failed to find convincing evidence of efficacy (i.e., $n > 1000$, $p < 10^{-6}$, $I^2 < 50\%$, absence of publication bias); publication bias was rarely assessed for the representative effect sizes. Eight effect sizes provided highly suggestive evidence (i.e., $n > 1000$, $p < 10^{-6}$), including smartphone interventions outperforming inactive controls on measures of psychological symptoms and quality of life ($d$s = 0.32 to 0.47) and text message-based interventions outperforming non-specific controls and active controls for smoking cessation ($d$s = 0.31 and 0.19, respectively). The magnitude of effects and strength of evidence tended to diminish as comparison conditions became more rigorous (i.e., inactive to active, non-specific to specific). Four effect sizes provided suggestive evidence, 14 effect sizes provided weak evidence, and eight effect sizes were non-significant. Despite substantial heterogeneity, no moderators were identified. Adverse effects were not reported. Taken together, results support the potential of mobile phone-based interventions and highlight key directions to guide providers, policy makers, clinical trialists, and meta-analysts working in this area.

The global burden of mental health continues to increase with illnesses like depression representing the single largest source of worldwide disability [1]. The World Health Organization estimates that this burden disproportionately impacts low-income countries [2], but even in

**Funding:** This research was supported by the National Center for Complementary and Integrative Health Grant K23AT010879 (S. B. Goldberg) and Grant K23AT011173 (S. Sun). J. Torous was supported by a grant from the Baer Foundation. O. Simonsson was supported by the Sweden-America Foundation. The funders had no role in study design, data collection and analysis, decision to publish, or preparation of the manuscript.

**Competing interests:** The authors have declared that no competing interests exist.

high income countries there is rising concern around unmet mental health needs, particularly during the COVID-19 pandemic [3]. While effective treatments for mental health disorders exist, including a variety of evidence-based therapies as well as medications–access to care is limited. For example, in a 2020 report the Substance Abuse and Mental Health Services Administration in the United States noted that to offer full access to evidence based mental health care in the United States would require training 4,486,865 new mental health professionals [4]. The stark reality of the simple impossibility of meeting rising needs with the current workforce or mental healthcare system, even in high income countries, has thus catalyzed interest in mobile mental health interventions.

While the global pandemic and COVID-19 have accelerated interest and uptake of mobile health interventions [5], the field has been active since smartphones became accessible to consumers. As early as 2012, smartphone apps were being studied for use in DBT treatment [6] and the New York times reported "therapy apps. . .may soon make psychological help accessible anytime, anywhere [7]." Interventions capitalizing on pre-smartphone mobile phone technology (e.g., text message-based interventions) have been studied even longer [8]. For the purposes of this review, we define mobile phone-based interventions as behavioral interventions delivered remotely through mobile phones. This can include a wide variety of approaches such as smartphone apps, text message-based interventions, apps integrated with wearable sensors such as Fitbits, as well as interventions that combine a mobile phone component with additional support (e.g., mobile phone-based intervention added to augment a clinician-delivered intervention). We do not consider interventions delivered through websites that could theoretically be access through smartphone but not specifically designed for mobile phones as mobile phone-based interventions. Likewise, interventions delivered by clinicians via videoconferencing or telephone (i.e., teletherapy) were not considered mobile phone-based interventions.

Mobile phone-based interventions may be particularly helpful for increasing access, as these devices are owned by the vast majority of the population and typically kept within arm's reach [9]. Today thousands of mental health apps are available for immediate download [10]. The landscape has expanded to such an extent that professional societies have created evaluation frameworks [11] and healthcare regulators around the world are exploring new ways to categorize and regulate this burgeoning space [12–14]. Large healthcare systems like Kaiser Permanente that had already integrated mobile health apps into care before COVID-19 doubled the number of app referrals in the first months of the pandemic (40,000 in May 2020 [15]), representing the continued expansion of mobile mental health interventions.

As interest and uptake of mobile phone-based intervention and mobile mental health interventions generally has increased, so has research on their efficacy. From fewer than five studies per year in 2011 [16] to now hundreds per year–there exist thousands of research studies on mobile health interventions. While the first generation of these studies focused on feasibility and acceptability, the accumulating evidence clearly indicates that people suffering from all mental health conditions, including even schizophrenia (which may be associated with severe disability that in theory could interfere with feasibility and acceptability), are interested in and willing to use technology as part of their care [17]. The newer generation of investigation now focuses on engagement, efficacy (i.e., performance in ideal conditions), and effectiveness (i.e., performance in naturalistic contexts) in ultimately seeking to answer questions around real world use of these interventions towards improving outcomes. Yet surprisingly, this research has not yielded clear answers with recent 2021 systematic reviews of app interventions reporting outcomes ranging from "inconsistent results" [18] to "proven effectiveness" [19].

Inconsistencies in the literature on mobile health interventions are also reflected in meta-analyses. Examining different portions of the literature have resulted in some meta-analysis of

mobile health apps to conclude that self-help apps "cannot be recommended" [20] while others that these apps may "serve as a cost-effective, easily accessible, and low intensity interventions" [21]. Meta-reviews (i.e., systematic reviews of meta-analyses) can be particularly helpful in instances like this [22]. Meta-reviews studies can clarify points of convergence and divergence within the meta-analytic literature and can more reliably guide providers and health care policy makers than a single meta-analysis. Meta-reviews can also clarify potential methodological shortcomings of both the meta-analytic and primary study literature to guide future work [22–24].

To our knowledge, Lecomte et al. [25] conducted the only meta-review investigating the effects of mobile phone-based interventions on mental health. They reviewed seven meta-analyses focused on mental health apps, concluding that apps for anxiety and depression hold clinical promise. While a valuable first assessment of this growing meta-analytic literature, the study has several important limitations which limit the strength of conclusions that can be drawn. One important limitation was the inclusion of both randomized and non-randomized studies, the latter of which cannot be used to draw causal inferences. In addition, Lecomte et al. [25] included effect size estimates that were based on the combination of active and inactive controls. Given robust evidence that the strength of the control condition impacts estimates of relative efficacy [26], such effect sizes provide ambiguous information regarding intervention effects. In addition, this review did not include text message-based interventions, which for some conditions (e.g., smoking) have a well-established evidence base[27]. Lastly, Lecomte et al. [25] did not identify a single effect size most likely to represent a particular outcome (e.g., effects on depression versus inactive controls). Lacking such a summary effect size may make findings less actionable for clinicians and other health care decision makers.

The varied conclusions of meta-analyses are understandable in the context of numerous use cases as well as schemas to classify mobile phone-based health interventions. Interventions focusing on prevention may target those without a diagnosed mental illness and offer different effects than those targeting acute or chronic illness management [28–29]. Interventions for depression may utilize a plethora of psychological techniques that each offer unique benefits to different clusters of patients. Studies reporting on the preliminary efficacy of mobile phone-based interventions may not require an active control group while those examining effectiveness (i.e., impact in the real world) may. Further complicating matters, clinical endpoints across studies are obfuscated by a panoply of self-reported outcomes and varied assessment schedules. Thus, each element in the participants, interventions, comparisons, outcomes [PICO] framework presents a potentially complex choice for any review or meta-analyses.

A further challenge in understanding the effect of mobile phone-based interventions is inconsistent reporting of methodology and outcomes in the literature itself. Results are rarely reported in terms of engagement, control group coding is inconsistent, testing of moderators infrequently completed, and publication bias not consistently assessed. These concerns are not strictly academic and in 2021 a pharmaceutical industry backed study of an app for schizophrenia reported negative results that the company ascribed to engagement and control group issues [30]. Thus, just as mobile health interventions for mental health have reached a peak of public interest with COVD-19, are entering into clinical care settings, and emerging as the focus of high stakes and high-cost clinical research studies–the strength of their evidence remains, and preferred meta-analytical methods, unclear.

With this framing, in the current study we conduct a meta-review of meta-analyses of mobile phone-based intervention tested in randomized controlled trials. We aim to clarify the strength of evidence across PICO categories (i.e., different pairings of types of participants, interventions, comparisons, and outcomes) and identify important study design considerations for future primary research and meta-analysis. Given the many ways this literature

could be examined (i.e., crossing of PICO) and an interest in clarifying points of convergence and divergence across existing meta-analyses, we utilize extant meta-analytic summaries to offer guidance to the public, clinicians, and researchers around mobile phone-based interventions. To do so, we followed an umbrella review methodology and identified representative effect sizes for specific PICO pairings that was based on the largest sample. Effect sizes were evaluated based on the certainty of the evidence using previously employed metrics [22,24].

## Method

### Protocol and registration

This meta-review was preregistered through the Open Science Framework (https://osf.io/s2t67/) and followed the Preferred Reporting Items for Systematic Reviews and Meta-analyses (PRISMA [31]) guidelines. We made three deviations from the protocol. First, we used a five-tier comparison condition coding scheme (inactive, active, non-specific, specific, adjunct). This allowed inclusion of effect sizes based on active comparison conditions that may or may not have been intended to be therapeutic (i.e., non-specific and specific) [32]. Second, we applied criteria drawn from umbrella reviews to evaluate certainty of evidence [22]. Third, we did not evaluate attrition as this was not reported across meta-analyses.

### Eligibility criteria

Studies were eligible if they (1) reported a meta-analytically derived effect size related to mobile phone-based interventions (2) on a mental health outcome (e.g., psychiatric symptoms, stress, quality of life, addictive behaviors that are included in the DSM-5 [33] such as alcohol and tobacco use) (3) based on a minimum of four randomized controlled trials (RCTs) [34] (4) using comparison conditions that could be categorized as inactive, active, non-specific, specific, or adjunct (i.e., added to an active treatment) [21]. We planned *a priori* to avoid combining across comparison condition types as these estimates are ambiguous to interpret and can lead to misleading results (e.g., interventions tested using more rigorous specific active control conditions appear less effective [35]). Effect sizes had to be reported in standardized units (e.g., Cohen's *d*, Hedges' *g*, odds ratio) along with a 95% confidence interval (CI) and be based on a sample of studies that did not combine comparison condition types (e.g., combined specific active controls and inactive controls as defined below). No restrictions were made based on other PICO categories. Interventions could include components beyond mobile phones (e.g., smartphones as adjunct intervention [21]) but must have included a mobile phone component (e.g., telephone-based interventions were not eligible).

### Information sources

We searched six databases (MEDLINE/PubMed, CINAHL, PsycINFO, Scopus, Web of Science, Cochrane Systematic Reviews). Databases were searched since inception until October 31st, 2020.

### Search

We used the following search terms: ("meta-analy*") AND ("smartphone*" OR "smart phone" OR "mobile phone" OR "cellular phone" OR "cell phone" OR "mobile app*" OR "mobile device" OR "mobile-based" OR "mobile health" OR "mhealth" OR "m-health" OR "iphone" OR "android" OR "tablet").

## Study selection

Titles and/or abstracts were independently and blindly screened for inclusion by pairs of two authors (SG, SL, OS, SS). Disagreements were discussed until reaching consensus. Interrater reliability was excellent (*kappa* $\geq$ .75) [36].

## Data collect process

All study-level data were independently coded using standardized spreadsheets with the exception of ratings of quality which were independently and blindly coded.

## Data items

Eligible effect sizes were extracted along with the associated 95% CI, the number of studies and participants represented, estimates of heterogeneity (i.e., $I^2$), and results of tests of publication bias (e.g., trim-and-fill, fail-safe N). We also coded results of moderator tests when these tests were conducted on an eligible effect size (i.e., not conducted across a sample combining comparison condition types). In order to facilitate summarizing across PICO subcategories, we coded sample population (e.g., adults, adolescents) and/or clinical condition (e.g., depression), intervention (e.g., text message, smartphone app), comparison condition, and outcome (e.g., depression, smoking cessation). Samples were considered clinical if the participants in the associated RCTs were diagnosed with a particular condition and/or reported elevated symptoms. To define clinical conditions, we followed definitions used in the eligible meta-analyses which included symptoms above a clinical threshold, a formal diagnosis of a specific disorder (e.g., depression) [20], or various indicators of smoking behavior [27, 37]. We coded interventions into six categories based on the groupings found within the eligible studies. Of note, these groupings were not mutually exclusive (i.e., a specific intervention could fall into multiple categories) but were created to reflect the groupings used in the eligible meta-analyses. These included smartphone apps (smartphone apps without additional support), smartphone-based interventions (smartphone apps with or without additional support), meditation apps (meditation apps with or without additional support), text message-based interventions (text messages with or without additional support), ecological momentary assessment (EMA) interventions (EMA with or without additional support), and mobile phone-based interventions which could include any combination of the previous categories. We used a five-tier scheme to separate comparison conditions into coherent subgroups. One commonly applied distinction was between inactive and active controls [38]. Conditions that involved no intervention beyond that received by the mobile phone intervention arm were coded as inactive. Waitlist, no treatment controls, and treatment-as-usual when the mobile phone arm also received this were all coded as inactive. Conditions that involved an active intervention were coded as active. Active interventions could include interventions that were designed to control for active components (such as time and attention) but not to provide therapeutic ingredients (e.g., attentional control) as well as interventions that were designed to provide therapeutic benefit (i.e., other interventions). However, instead of separating comparison conditions based on whether they were active or inactive, some meta-analyses [39] separated comparison conditions based on whether they were intended to be therapeutic [32]. Thus, effect sizes based on the combination of inactive conditions (e.g., waitlist) and active conditions not intended to be therapeutic (e.g., attentional control) were coded as non-specific controls. In other words, no specific intervention ingredients intended to be therapeutic were included. Comparison conditions that were restricted to active interventions that were intended to be therapeutic (i.e., other therapies) were coded as specific active controls. A final category included comparisons

between active interventions with and without a mobile phone-based intervention added (i.e., adjunct intervention) [21].

When available in the included meta-analyses, we coded evaluations of primary study risk of bias (e.g., Cochrane risk of bias tool [40]) and reports of adverse events. We coded the quality of each meta-analysis using the National Institutes of Health (NIH) Quality Assessment of Systematic Reviews and Meta-Analyses Tool [41] and interpreted scores in line with previous meta-reviews [42] where 7 or 8 indicates "good," 4 to 6 "fair," and 0 to 3 "poor" quality. In order to describe the magnitude of the primary study literature, we coded each primary RCT's sample size, country, and year of publication.

## Summary measures

Standardized mean difference (i.e., Cohen's *d*, Hedges' *g*) served as our primary effect size measure. As both effect sizes are in standardized mean difference units (with Hedges' *g* accounting for small sample bias [43]), we refer to them together as Cohen's *d*. Alternative effect size measures (e.g., odds ratio, hazard ratio) were converted into Cohen's *d* using standard methods [44]. The magnitude of Cohen's *d* and $I^2$ were interpreted using established guidelines [45, 46].

We evaluated the certainty of the evidence using criteria drawn from umbrella review methodology [22, 24]. Convincing evidence required: $n > 1000$, $p < 10^{-6}$, no evidence of publication bias, low to moderate heterogeneity ($I^2 < 50\%$). Highly suggestive evidence required: $n > 1000$, $p < 10^{-6}$. Suggestive evidence required: $n > 1000$, $p < 10^{-3}$. Weak evidence required: $p < .050$. *P*-values were calculated from confidence intervals using standard methods [47].

## Synthesis of results

For each PICO subcategory, we identified a representative effect size that was based on the largest sample, which in theory would provide the most reliable estimate. The specific PICO subcategories were identified inductively based on categories utilized in the available meta-analyses. When multiple effect sizes within a given PICO subcategory were reported (e.g., effects of text messaging on depression symptoms, effects of text messaging on Beck Depression Inventory [48]), we selected the effect size based on the larger sample. We organized our reporting of results by outcome domains, reviewing effect size magnitude and certainty of the evidence separated by population, intervention, and comparison (i.e., the remaining PICO).

# Results

## Study selection

A total of 4,447 citations were retrieved (S1 Fig), with 14 meta-analyses reporting eligible effect sizes. Thirty-six potentially eligible meta-analyses were excluded due to combining either inactive and active controls or non-specific and specific controls. The 14 meta-analyses represented data from 145 unique primary RCTs with 47,940 participants.

## Study characteristics

Meta-analysis-level characteristics are shown in Table 1. Meta-analyses included an average of 18.71 studies (SD = 16.22, range = 6 to 66). In terms of population, six meta-analyses (42.9%) were focused exclusively on adults while eight (57.1%) included studies from both adult and adolescent/adult samples. The most commonly investigated clinical condition was smoking (*k* = 5, 35.7%), with two meta-analyses (14.3%) focusing on individuals with elevated mental

**Table 1. Characteristics of included meta-analyses.**

| Meta-analysis | Population | Condition | Intervention | Outcomes | k | RoB | NIH |
|---|---|---|---|---|---|---|---|
| Cox (2020) [48] | adults | n/a | text messaging | depression | 9 | Cochrane, GRADE | 7 |
| Do (2018) [70] | adults/ adolescents | smoking | text messaging | smoking cessation | 6 | Cochrane | 8 |
| Firth (2017a) [38] | adults | mental health concerns | smartphone intervention | anxiety | 9 | Cochrane | 6 |
| Firth (2017b) [50] | adults | n/a | smartphone intervention | depression | 18 | Cochrane | 7 |
| Gál (2021) [39] | adults | n/a | meditation apps | anxiety, depression, stress, wellbeing | 34 | Cochrane | 7 |
| Gee (2016) [49] | adults/ adolescents | n/a | ecological momentary interventions | anxiety | 6 | Cochrane | 8 |
| Linardon (2019) [21] | adults/ adolescents | n/a | smartphone intervention | depression, anxiety, stress, quality of life | 66 | Cochrane | 6 |
| Linardon (2020) [71] | adults/ adolescents | n/a | smartphone app | self-compassion, mindfulness/acceptance, depression/distress | 33 | Cochrane | 7 |
| Scott-Sheldon (2016) [51] | adults | smoking | text messaging | smoking cessation | 16 | Jadad and other measures | 7 |
| Senanayake (2019) [72] | adults/ adolescents | depression | text messaging | depression | 7 | Joanna Briggs Institute | 7 |
| Spohr (2015) [52] | adults/ adolescents | smoking | text messaging | smoking cessation | 13 | n/a | 7 |
| Weisel (2019) [20] | adults | mental health concerns | smartphone app | depression, anxiety, suicidal ideation, smoking/drinking | 16 | n/a | 7 |
| Whittaker (2016) [37] | adults/ adolescents | smoking | text messaging / smartphone app | smoking cessation | 12 | Cochrane, GRADE | 7 |
| Whittaker (2019) [27] | adults/ adolescents | smoking | text messaging / smartphone app | smoking cessation | 17 | Cochrane, GRADE | 8 |

Note: k = number of included studies; ROB = risk of bias assessment method; NIH = National Institutes of Health Quality Assessment of Systematic Reviews and Meta-Analyses Tool. n/a = not applicable (i.e., not clinical condition required for inclusion).

health symptoms, and one (7.1%) focusing on individuals with depression. Average meta-analysis quality was 7.07 out of 8 (SD = 0.62, range = 6 to 8). All but two meta-analyses received a 7 or 8, indicating "good" quality. Meta-analyses were published between 2015 and 2020. All studies with the exception of those focused on smoking assessed outcomes at post-treatment. Smoking studies commonly included longer-term follow-up assessment (6 or 12 months post-quit attempt [27, 37]).

Primary studies represented in the 14 meta-analyses had an average sample size of 330.62 (SD = 747.73, range = 8 to 5,800). Studies occurred in North America (43.4%), Europe (35.2%), Oceania (Australia or New Zealand; 10.3%), Asia (6.9%), the Middle East (2.1%), or across multiple regions (2.1%). Primary studies were published between 2005 and 2020 with a median year of 2017.

## Risk of bias within studies

Most meta-analyses evaluated risk of bias (k = 12, 85.7%), most commonly using the Cochrane tool (k = 10, 71.4%). Three studies used GRADE (21.4%). Fig 1 displays a summary of bias assessment. Blinding of personnel and participants was the area most consistently rated as high risk for bias (44.5%) and incomplete outcome data was the second most common (26.5%). There was evidence for some inconsistency in the coding of bias, specifically in the domain of blinding (S1 Table; S2 Fig). Several meta-analyses indicated high risk of bias related to blinding of personnel and participants, while also indicating that outcome assessors were

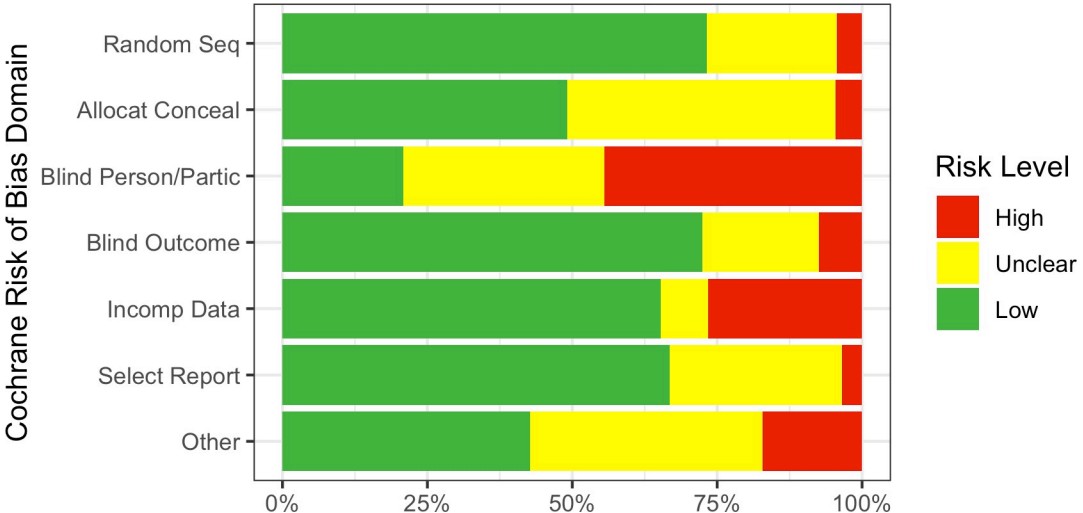

**Fig 1. Risk of bias summary aggregated across meta-analyses.** Random Seq = random sequence generation; Allocat Conceal = allocation concealment; Blind Person/Partic = blinding of personnel and participants; Blind Outcome = blinding of outcome assessor; Incomp Data = incomplete outcome data; Select Report = selective reporting; Other = other bias.

blind [48]. It appeared that authors may have considered a lack of interaction with study personnel as reflecting low risk of bias for outcome assessors when outcomes were self-reported [39].

Although many of the included meta-analyses assessed publication bias, this was often done using the full sample of studies (i.e., not separated by comparison condition type). Publication bias was tested for only two of the 34 eligible effect size estimates. Of these, one indicated evidence for publication bias (upwardly biased original estimate [49]) and one did not [50].

## Results of individual studies

After removing non-representative effect sizes (i.e., effect sizes that overlapped with another effect size with equivalent PICO based on a larger sample), 34 unique effect sizes were retained. Table 2 displays representative effect sizes which uniquely capture a PICO combination along with grading of the certainty of the evidence. Effect sizes were based on an average of 9.94 RCTs (SD = 8.29, range = 4 to 37) and 2,707 participants (SD = 4,374, range = 246 to 19,368). In terms of intervention, 15 effect sizes (44.1%) were from studies investigating smartphone interventions (e.g., apps with or without additional support), seven (20.6%) from smartphone apps, six (17.6%) from meditation apps, four (11.8%) from text messaging, one (2.9%) from ecological momentary interventions, and one (2.9%) from mobile phone-based interventions (i.e., combination of text messaging and smartphone interventions). In terms of comparison condition, nine (26.5%) were from comparisons with inactive controls, 13 (38.2%) from comparisons with non-specific controls, five (14.7%) from comparisons with active controls, five (14.7%) from comparisons with specific active controls, and two (5.9%) from comparisons between active treatments with and without an adjunctive mobile phone-based intervention.

## Synthesis of results

Across outcome categories, 11 effect sizes were related to anxiety (32.4%), 10 to depression (29.4%), four to smoking (11.8%), three to stress (8.8%), three to quality of life (8.8%), one to

**Table 2. Representative effect sizes across PICO categories.**

| Out | Pop | Cond | Intervention | Comp | Meta-analysis | k | n | ES | CI | I² | Pub | Strength |
|-----|-----|------|--------------|------|---------------|---|---|-----|-----|-----|-----|----------|
| Anx | adult | n/a | smartphone | inactive | Linardon (2019) | 28 | 3,093 | 0.32 | [0.19, 0.44] | 63 | n/a | high suggest |
| Anx | adult | ↑ sx | smartphone | inactive | Firth (2017a) | 6 | 1,212 | 0.45 | [0.30, 0.61] | 32.4 | n/a | high suggest |
| Anx | adult | ↑ sx | app | inactive | Weisel (2019) | 6 | 806 | 0.49 | [0.27, 0.71] | 47 | n/a | weak |
| Anx | adult | n/a | med app | inactive | Gál (2021) | 10 | 1,381 | 0.31 | [0.17, 0.46] | 48 | n/a | suggestive |
| Anx | mix | n/a | EMA | non-specific | Gee (2016) | 6 | 1,021 | 0.31 | [0.07, 0.55] | 17.78 | yes | weak |
| Anx | adult | ↑ sx | app | non-specific | Weisel (2019) | 8 | 948 | 0.43 | [0.19, 0.66] | 66 | n/a | weak |
| Anx | adult | anxious | app | non-specific | Weisel (2019) | 4 | 479 | 0.3 | [-0.10, 0.70] | 75 | n/a | non-sig |
| Anx | adult | ↑ sx | smartphone | active | Firth (2017a) | 5 | 1,026 | 0.19 | [0.07, 0.31] | 0 | n/a | weak |
| Anx | adult | n/a | smartphone | active | Linardon (2019) | 8 | 890 | 0.18 | [0.07, 0.29] | 7 | n/a | weak |
| Anx | adult | n/a | med app | specific | Gál (2021) | 4 | 337 | 0.26 | [-0.00, 0.52] | 0 | n/a | non-sig |
| Anx | adult | n/a | smartphone | specific | Linardon (2019) | 4 | 246 | 0.09 | [-0.21, 0.39] | 32 | n/a | non-sig |
| Dep | adult | n/a | med app | inactive | Gál (2021) | 8 | n/a | 0.35 | [0.24, 0.47] | 9 | n/a | weak |
| Dep | adult | n/a | smartphone | inactive | Linardon (2019) | 34 | 3,907 | 0.32 | [0.22, 0.42] | 52 | n/a | high suggest |
| Dep | adult | n/a | text | non-specific | Cox (2020) | 9 | 1,918 | 0.27 | [0.00, 0.54] | 82.5 | n/a | weak |
| Dep | mix | n/a | smartphone | non-specific | Linardon (2019) | 8 | 1,840 | 0.39 | [0.21, 0.58] | 60 | n/a | suggestive |
| Dep | adult | ↑ sx | app | non-specific | Weisel (2019) | 12 | 1,544 | 0.34 | [0.18, 0.49] | 53 | n/a | suggestive |
| Dep | adult | depressed | app | non-specific | Weisel (2019) | 6 | 796 | 0.33 | [0.10, 0.57] | 59 | n/a | weak |
| Dep | adult | n/a | smartphone | active | Firth (2017b) | 12 | 2,381 | 0.22 | [0.10, 0.33] | 47.2 | no | suggestive |
| Dep | adult | n/a | med app | specific | Gál (2021) | 5 | 981 | 0.28 | [0.09, 0.48] | 0 | n/a | weak |
| Dep | adult | n/a | smartphone | specific | Linardon (2019) | 12 | 751 | 0.13 | [-0.07, 0.34] | 60 | n/a | non-sig |
| Dep | adult | n/a | smartphone | adjunct | Linardon (2019) | 4 | n/a | 0.26 | [-0.09, 0.61] | 71 | n/a | non-sig |
| Smoke | mix | smokers | mobile | non-specific | Whittaker (2016) | 12 | 11,885 | 0.3 | [0.22, 0.38] | 59 | n/a | high suggest |
| Smoke | mix | smokers | text | non-specific | Whittaker (2019) | 13 | 14,133 | 0.31 | [0.24, 0.38] | 71 | n/a | high suggest |
| Smoke | adult | smokers | text | active | Scott-Sheldon (2016) | 16 | 19,364 | 0.19 | [0.14, 0.24] | n/a | n/a | high suggest |
| Smoke | adult | smokers | text | adjunct | Whittaker (2019) | 4 | 997 | 0.31 | [0.08, 0.54] | 0 | n/a | weak |
| SU | adult | ↑ sx | app | non-specific | Weisel (2019) | 5 | 1,732 | 0.18 | [-0.09, 0.45] | 81 | n/a | non-sig |
| Stress | adult | n/a | smartphone | inactive | Linardon (2019) | 20 | 2,558 | 0.47 | [0.33, 0.62] | 60 | n/a | high suggest |
| Stress | adult | n/a | med app | inactive | Gál (2021) | 8 | 923 | 0.62 | [0.24, 1.01] | 80 | n/a | weak |
| Stress | adult | n/a | smartphone | active | Linardon (2019) | 6 | 929 | 0.09 | [-0.05, 0.24] | 0 | n/a | non-sig |
| SI | adult | ↑ sx | app | non-specific | Weisel (2019) | 4 | 286 | 0.14 | [-0.10, 0.37] | 0 | n/a | non-sig |
| QOL | mix | n/a | smartphone | inactive | Linardon (2019) | 37 | 4,672 | 0.35 | [0.28, 0.43] | 29 | n/a | high suggest |
| QOL | adult | n/a | smartphone | non-specific | Linardon (2019) | 4 | 489 | 0.41 | [0.21, 0.61] | 0 | n/a | weak |
| QOL | adult | n/a | smartphone | specific | Linardon (2019) | 6 | 388 | 0.02 | [-0.14, 0.17] | 0 | n/a | non-sig |
| WB | adult | n/a | med app | non-specific | Gál (2021) | 4 | n/a | 0.31 | [0.05, 0.56] | 0 | n/a | weak |

Note: PICO = participants, interventions, comparisons, outcomes; Out = outcome; Pop = population; Cond = condition; Comp = comparison; k = number of studies; n = sample size; ES = effect size in Cohen's *d* units; CI = 95% confidence interval; I² = heterogeneity estimate; Pub = indication of publication bias (coded as n/a if not reported); Strength = evaluation of evidence strength; Anx = anxiety; Dep = depression; Smoke = smoking cessation; SU = substance use (smoking/drinking); SI = suicidal ideation; QOL = quality of life; WB = wellbeing; adult = adult sample; mix = mixture of adult and adolescent samples; ↑ sx = elevated symptoms; smartphone = smartphone-based interventions (smartphone apps with or without additional support); app = smartphone app without additional support; med app = meditation app with or without additional support; EMA = ecological momentary assessment intervention with or without additional support; text = text message-based intervention with or without additional support; mobile = mobile phone-based interventions which could include any combination of mobile phone-based intervention types; inactive = control conditions without active component (e.g., waitlist); active = control conditions that included an active component that may or may not have been intended to be therapeutic; non-specific = non-specific controls which included inactive condition and active conditions that were not intended to be therapeutic; specific = specific active controls which included active controls that were intended to be therapeutic; adjunct = comparison between active interventions with and without a mobile phone-based intervention added; high suggest = highly suggestive evidence ($n > 1000$, $p < 10^{-6}$); suggest = suggestive evidence ($n > 1000$, $p < 10^{-3}$); weak = weak evidence ($p < .050$); non-sig = non-significant effect ($p > .050$).

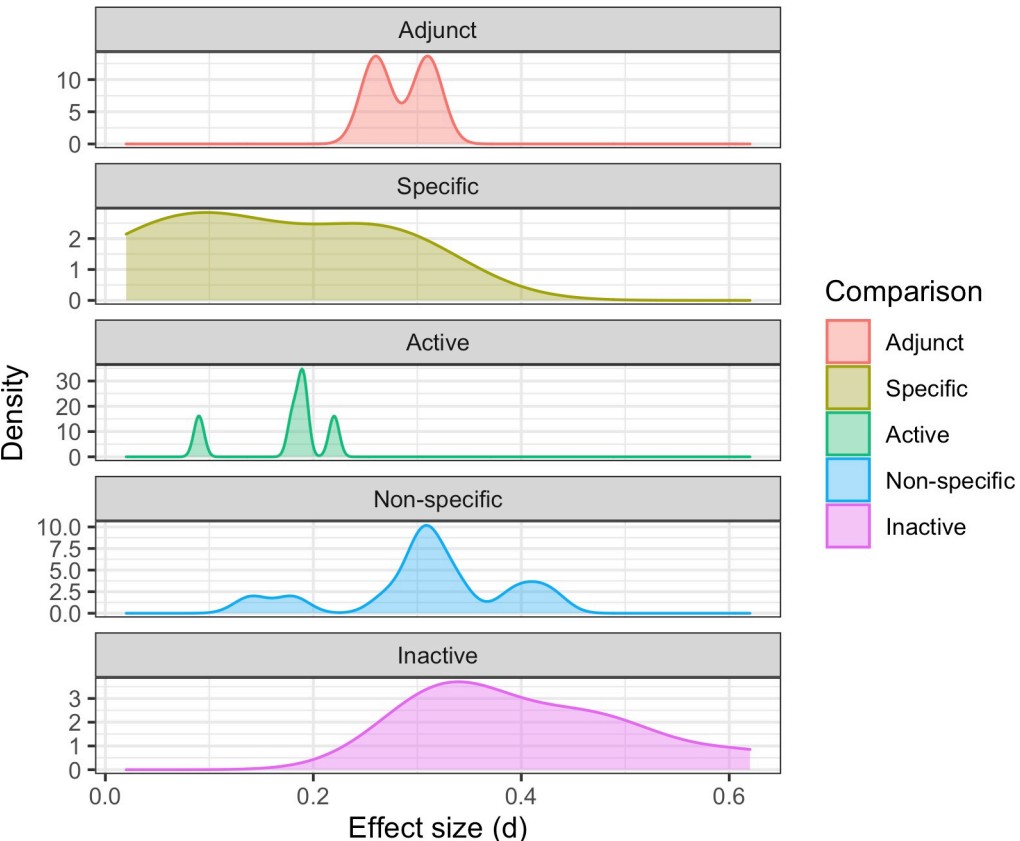

**Fig 2. Density plots displaying distribution of representative effect sizes separated by comparison type.**
Inactive = no active comparison (e.g., assessment only, waitlist control); Non-specific = non-specific controls (i.e., not intended to be therapeutic); Active = active comparison that may or may not have included therapeutic ingredients; Specific = specific active controls (i.e., intended to be therapeutic); Adjunct = mobile phone-based intervention tested as adjunct to another intervention.

wellbeing (2.9%), one to suicidal ideation (2.9%), and one to smoking/drinking (2.9%). Sixteen (47.1%) of the representative effect sizes were based on $n > 1000$. Nine (26.5%) were significant at $p < 10^{-6}$, 16 (47.1%) were significant at $p < 10^{-3}$, and 25 (73.5%) were significant at $p < .050$. Average heterogeneity was 38.24% (SD = 30.06, range = 0 to 82.5) and sixteen (47.1%) had low to moderate heterogeneity ($I^2 < 50\%$). Evidence was graded as highly suggestive for eight effect sizes (23.5%), as suggestive for four effect sizes (11.8%), and as weak for 14 effect sizes (38.2%). Eight effect sizes (26.5%) were non-significant. No effect size was graded as convincing.

Figs 2 and 3 display the distribution of effect sizes separated by comparison. Both figures illustrate heterogeneity across estimates within a given comparison type, but a generally monotonic movement of effect sizes towards zero as the comparison condition becomes more rigorous (i.e., moving from inactive to specific active). Fig 4 displays the distribution of effect sizes separated by outcome domain.

**Anxiety.** Compared to inactive controls, smartphone interventions showed highly suggestive evidence of small magnitude effects on anxiety in the general population ($d = 0.32$ [21]) and among those with elevated symptoms ($d = 0.45$ [38]). Compared to inactive controls, meditation apps showed suggestive evidence of small magnitude effects ($d = 0.31$ [39]). There was weak evidence of small magnitude effects of apps compared to inactive controls among those with elevated symptoms ($d = 0.49$ [20]), downgraded from highly suggestive due to the small

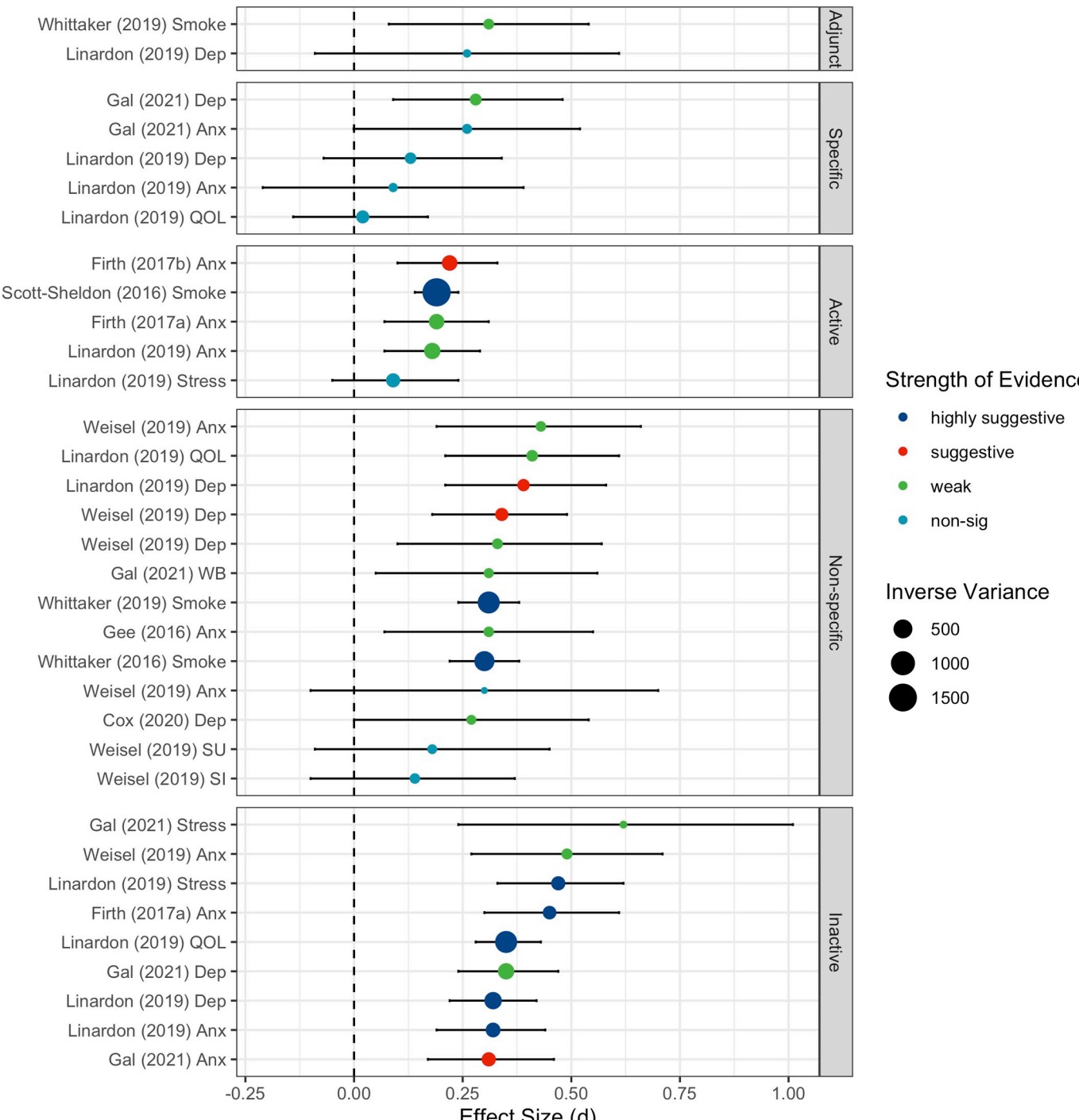

**Fig 3. Forest plot of representative effect sizes separated by comparison type.** Color based on strength of evidence and size based on inverse variance. Non-specific = non-specific controls (i.e., not intended to be therapeutic); Specific = specific active controls (i.e., intended to be therapeutic); Adjunct = mobile phone-based intervention tested as adjunct to an active treatment.

sample size ($n$ = 806). Evidence for ecological momentary interventions and apps compared to non-specific controls was weak or non-significant, although of similar magnitude ($d$s = 0.30 to 0.43 [20, 49]). Smartphone interventions showed weak evidence of very small effects compared

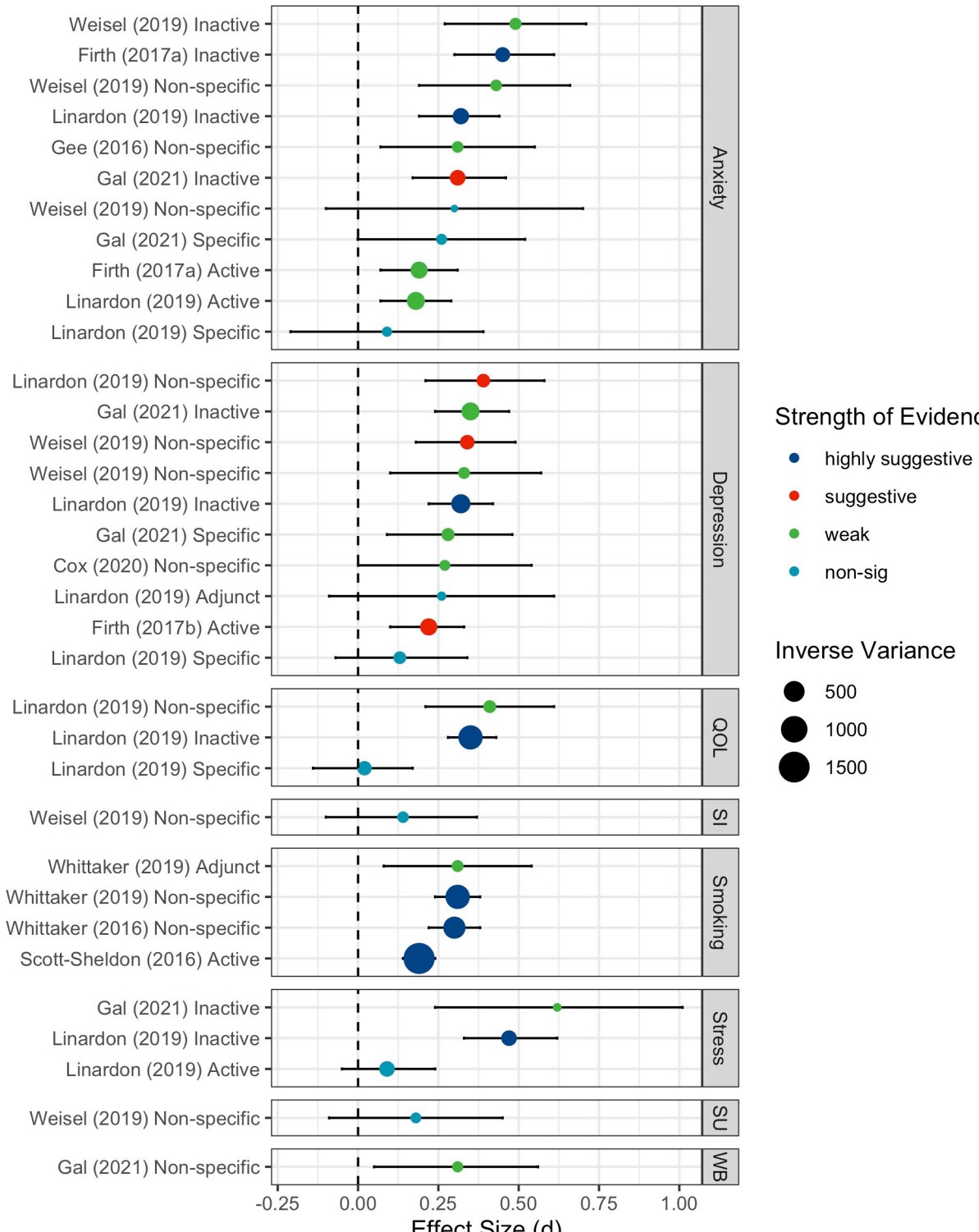

**Fig 4. Forest plot of representative effect sizes separated by outcome domain.** Color based on strength of evidence and size based on inverse variance. Non-specific = non-specific controls (i.e., not intended to be therapeutic); Specific = specific active controls (i.e., intended to be therapeutic); Adjunct = mobile phone-based intervention tested as adjunct to another intervention.

to active controls in the general population and among those with elevated symptoms (*d*s = 0.18 and 0.19, respectively [21, 38]). Compared to specific active controls, meditation apps and smartphone interventions were not significantly superior in their effects on anxiety (*d*s = 0.26 and 0.09, respectively [21, 39]).

**Depression.** Compared to inactive controls, smartphone interventions showed highly suggestive evidence of small magnitude effects on depression (*d* = 0.32 [21]) and meditation apps showed weak evidence of small magnitude effects (*d* = 0.35 [39]). Compared to non-specific controls, smartphone interventions showed suggestive evidence of small magnitude effects (*d* = 0.39 [21]) as did app interventions among those with elevated symptoms (*d* = 0.34 [20]). Evidence was weak for small effects of app interventions versus non-specific controls among those with depression (*d* = 0.33 [20]) and for text message-based interventions (*d* = 0.27 [48]). Smartphone apps showed suggestive evidence of small effects compared to active controls (*d* = 0.22 [50]). Meditation apps showed weak evidence of small effects compared to specific active controls (*d* = 0.28 [39]). Smartphone interventions did not differ significantly from specific active controls (*d* = 0.13 [21]) or when tested as an adjunct to treatment (*d* = 0.26 [21]).

**Smoking and smoking/drinking.** Compared to non-specific controls, both mobile phone interventions and text message-based interventions showed highly suggestive evidence of small magnitude effects on smoking cessation (*d*s = 0.30 to 0.31 [27, 37]). Text message-based interventions showed highly suggestive evidence of very small magnitude effects compared to active controls on smoking cessation (*d* = 0.19 [51]) and weak evidence of small magnitude effects when tested as an adjunct to other support on smoking cessation (*d* = 0.31 [27]). Apps did not differ from non-specific controls on a combination of smoking and drinking outcomes (*d* = 0.18 [20]).

**Stress.** Compared to inactive controls, smartphone interventions showed highly suggestive evidence of small magnitude effects on stress (*d* = 0.47 [21]). Meditation apps showed weak evidence of moderate magnitude effects compared to inactive controls (*d* = 0.62 [39]). Smartphone interventions did not differ from active controls (*d* = 0.09 [21]).

**Suicidal ideation.** Only one effect size characterized effects on suicidal ideation. Apps did not differ from non-specific controls (*d* = 0.14 [20]).

**Quality of life and wellbeing.** Compared to inactive controls, smartphone interventions showed highly suggestive evidence of small magnitude effects on quality of life (*d* = 0.35 [21]). Compared to non-specific controls, smartphone interventions showed weak evidence of small magnitude effects on quality of life (*d* = 0.41 [21]) and meditation apps showed weak evidence of small magnitude effects on wellbeing (*d* = 0.31 [39]). Smartphone interventions did not differ from specific active controls on quality of life (*d* = 0.02 [21]).

## Moderators

Only one study tested moderators within an eligible effect size (i.e., not using a sample that combined across comparison types). Spohr et al. [52] tested nine moderators as predictors of smoking cessation in text message-based interventions. These included study design features (follow-up length) and aspects of the intervention (e.g., text message only, message frequency, message type, inclusion of social support, inclusion of nicotine replacement therapy). None of these features significantly moderated treatment effects.

## Adverse events

Only one meta-analysis evaluated adverse events. Cox et al. [48] indicated that adverse events were not reported in the seven trials they included testing text message-based interventions for depression.

## Discussion

Psychological interventions delivered via mobile technology have been proposed as a means for reducing the global burden of disease associated with mental illness [53]. We sought to rigorously summarize and evaluate the strength of the available empirical evidence by conducting a meta-review of meta-analyses of mobile phone-based interventions tested in RCTs. Testament to the dramatic growth in this literature, the 14 meta-analyses we included were comprised of 145 primary studies representing 47,940 participants published since 2005 with 2017 being the median year of publication.

Applying standards drawn from umbrella review methodology [22, 24], we failed to find convincing evidence in support of any mobile phone-based intervention on any outcome. One reason evidence could not be graded as convincing was the lack of publication bias assessment within the meta-analyses themselves necessary for ruling out influence due to small sample bias. This highlights the importance of future meta-analyses including publication bias assessment at the effect size-level (i.e., not across the full sample if studies used differing comparison types).

Eight of the 34 representative effect sizes evaluated were graded as providing highly suggestive evidence, based on having large sample ($n > 1000$) and low $p$-value ($p < 10^{-6}$). Specifically, smartphone interventions outperformed inactive controls on measures of psychological symptoms (anxiety, depression, stress) and quality of life with small magnitude effects ($d$s = 0.32 to 0.47). Mobile phone-based interventions and text message-based interventions outperformed non-specific controls (e.g., attentional controls) on smoking cessation ($d$s = 0.30 and 0.31, respectively). The only comparison with an active control condition that yielded highly suggestive evidence was text message-based interventions for smoking cessation ($d$ = 0.19). While none of the highly suggestive effect sizes were moderate or larger in magnitude, they can be taken as proof-of-concept evidence that mobile phone-based interventions can at least modestly reduce some psychological symptoms and smoking. Scaled at a population level, even small effects may meaningfully impact public health [54].

Across the literature, we saw a general pattern of weakening evidence (i.e., fewer effect sizes graded as highly suggestive or suggestive) and diminishing effect sizes as the comparison condition became more rigorous. This finding is consistent with the broader psychotherapy literature which has highlighted the importance of the comparison condition when designing and interpreting the results of RCTs (e.g., whether determining absolute vs. relative efficacy [26, 55]). Three effect sizes indicated suggestive evidence (i.e., $n > 1000$ and $p < 10^{-3}$) of small magnitude superiority for mobile phone-based interventions (smartphone interventions, apps, meditation apps) on psychological symptoms (depression, anxiety) relative to inactive controls or non-specific controls (i.e., combination of inactive controls and active controls not intended to be therapeutic; $d$s = 0.31 to 0.39). In addition, we found suggestive evidence that smartphone interventions produce small magnitude effects relative to active controls on depression ($d$ = 0.22).

Thirteen of the 34 effect sizes provided weak ($p < .050$) evidence for mobile phone-based interventions. Several effect sizes were downgraded from suggestive to weak due to limited sample size ($n < 1000$). One notable effect size categorized as providing weak evidence was the comparisons between meditation apps and specific active controls (i.e., other interventions intended to be therapeutic) on depression ($d$ = 0.28 [39]). As a point of comparison, the upper bound of the effect size for the difference between various forms of psychotherapy is estimated to be $d$ = 0.20 [26]. Thus, given the rigorous comparison condition, should this effect size which is slightly larger than 0.20 persist as the literature grows and demonstrate robustness to publication bias, it may indicate an instance in which mobile phone-based interventions are particularly promising.

All of the remaining comparisons with specific active controls failed to demonstrate superiority of mobile phone-based interventions. Similarly, smartphone interventions did not yield benefits on depression when added as adjunctive to other treatments, although the effect was of small magnitude ($d$ = 0.26 [21]).

Taken together, these results suggest that mobile phone-based interventions may hold promise for modestly reducing common psychological symptoms (e.g., depression, anxiety), although effect sizes are generally small and rarely do these interventions outperform other interventions intended to be therapeutic (i.e., specific active controls). Text message-based interventions appear particularly effective in supporting smoking cessation. Despite modest effect sizes, the relatively low cost and high scalability of most of the interventions tested supports their public health relevance. Pragmatic RCTs and dissemination and implementation research will be crucial for evaluating the degree to which efficacy findings translate into real world effectiveness. This is especially important given the rapid dropout of user engagement for some forms of mobile phone interventions (e.g., health apps generally and mental health apps specifically [56, 57]). Determining safety of these interventions is also essential; discussion of adverse events was almost entirely absent from the meta-analytic literature. While limited assessment of adverse events is an issue in the broader psychotherapy literature [58], it may be particularly problematic to ignore within the context of mobile phone-based interventions which often include less support than traditional treatments. Unfortunately, to date is appears content for managing safety-related crises (e.g., suicidality) is not included in the majority of mental health apps [59].

There are several limitations necessary to consider when evaluating these findings. As is always the case with meta-analyses and meta-reviews, our study was limited by the available meta-analytic and primary study literature. It may well be that a body of literature exists pertaining the mobile phone-based interventions effects on mental health outcomes that was simply not meta-analyzed (or was meta-analyzed in a way that combined comparison conditions). Likewise, it is possible that the strength of the evidence may have been underestimated due to lack of publication bias assessment. In addition, the heterogeneity evident both within and between meta-analyses decreases confidence in any particular point estimate and highlights the potential of systematic differences in efficacy. Unfortunately, valid tests of moderators that might explain variability within a specific meta-analysis were rare and the one meta-analysis testing moderators within an eligible effect size found no significant predictors [52]. Thus, despite almost half of the representative effect sizes showing moderate heterogeneity ($I^2 >$ 50%), the included meta-analyses provided no clear indication of features that may account for this variability. Similarly, although there was a generally monotonic pattern of decreasing effects as the comparison condition became more rigorous, there was also substantial variability between representative effect sizes even within a comparison condition type. This suggests mobile phone-based interventions may vary in efficacy across PICO (e.g., based on participants, interventions, comparisons, outcomes). Another possibility not directly evaluated within the included meta-analyses is that quality of mobile phone-based interventions varies in important ways. The included primary studies tested a variety of interventions, some of which are widely used commercial products (e.g., Headspace [60, 61]) while others are interventions designed in collaboration with researchers (e.g., Wildflowers [62]). Lacking standardized assessment of intervention quality and related constructs (e.g., usability, acceptability, engagement), it is difficult for meta-analysts to evaluate whether effect sizes varied due to characteristics of the mobile phone-based interventions themselves.

Confidence in these results is also diminished by indication of risk of bias associated with lack of blinding of personnel and participants as well as incomplete outcome data. Many authors considered the included studies of low risk for bias due to blinding of outcome

assessors. However, the widespread reliance on self-report measures within this literature coupled with the lack of participants blinding and difficulty inherent in blinding participants to behavioral interventions [26] raises questions regarding the degree to which outcome assessors are in fact blind.

These limitations notwithstanding, the current study highlights several important future directions for both meta-analyses and primary studies. One potentially rich area may be the application of text message-based interventions for addictive behaviors [63]. While our review included several meta-analyses investigating text messaging for smoking cessation, only one effect size included assessment of other addictive behaviors (smoking and drinking combined [20]). Other PICOs that were relatively underrepresented in the meta-review include comparisons testing active interventions with and without a mobile phone-based adjunct, comparisons with specific active controls, and comparisons focused on non-adult samples. It is not possible to say from the current study whether these omissions are due to limited primary RCTs or meta-analyses of the available RCTs. Nonetheless, they may be important areas for future work of both kinds. In addition, we hope our results sensitize both clinical trialists and meta-analysts to the importance of considering the comparison condition (for an exemplary meta-analytic treatment of comparison type, see Linardon et al. [21]). This literature would be strengthened through more studies including active and ideally specific active controls which are capable of identifying key intervention ingredients and disentangling intervention-specific elements from the effect of expectancy and other non-specific factors alone (although these non-specific elements are likely an important component worthy of study in their own right [64, 65]). The quality of the primary study literature would be improved through the use of objective measures (to reduce bias due to unblinded outcome assessors), use of intention-to-treat analyses (to reduce bias due to incomplete outcome data), and preregistration of outcomes (to reduce selective reporting bias). A crucial future direction is consistent assessment and reporting of adverse events within both the primary studies and the meta-analytic literature. Surprisingly, only one meta-analysis mentioned adverse events, indicating the primary studies did not report on adverse events. Inconsistent assessment and evaluation of potential harm is a widespread issue within the psychotherapy literature [58], and is an important understudied area within mobile health research. As noted, the efficacy of mobile phone-based interventions among youth and adolescents also appears to be an understudied area and no eligible effect sizes were focused exclusively on this population. Future primary studies with youth and adolescent samples may be warranted, particularly given evidence of clinical need, acceptability, and potential efficacy of mobile health interventions for this group [66].

Future meta-analysts could consider grading the strength of their meta-analytic evidence using umbrella review methods. Given evidence that effects vary based on comparison condition, moderators will ideally be tested within a subsample of studies sharing a comparison condition type. Moderator tests are an important method to support efforts determining which intervention components appear most efficacious. Candidate moderators might include the degree of research staff interaction or therapist support, app quality [21], and whether an intervention was designed to prevent versus treat symptoms. It may also be informative to test whether the therapeutic model included in a given mobile phone-based intervention (e.g., cognitive behavioral therapy vs. mindfulness) moderates effects. Aggregating patient-level data across studies (i.e., individual patient meta-analysis) would be another powerful method for identifying moderators [67]. It would also be valuable to closely examine the effects of mobile phone-based interventions at various follow-up timepoints (e.g., 2-months, 6-months, 12-months post-baseline). Unlike the psychotherapy literature in which treatments are often time-limited and meta-analyses can cleanly examine effects at post-treatment versus follow-up [68], mobile phone-based interventions particularly those without guidance can be easily

accessed ongoingly thus making demarcation of "post-treatment" more ambiguous. However, a future meta-analysis might examine the persistence of effects at varying distances from baseline by including this characteristic as a moderator and/or assessing effects restricted to those measured within certain timeframes (as is commonly done in Cochrane Reviews [69]).

Lastly, the current findings have public health and health policy implications. While failing to demonstrate convincing evidence, the highly suggestive evidence for some mobile phone-based interventions on some outcomes (e.g., smartphone interventions on depression, anxiety, and stress; text message interventions on smoking cessation) supports future research in this area as well as consideration of these approaches as cost-effective means for reducing common psychiatric symptoms and supporting smoking cessation. Mobile phone-based interventions may be worth considering as prevention tools, or as initial interventions within a stepped care model. They may also serve as useful adjunct to traditional treatment, although we found only weak evidence supporting this possibility (text message-based interventions for smoking cessation [27]). Eventually, standardized and transparent formal evaluation of these interventions' clinical efficacy (e.g., by the United States Food and Drug Administration) may help guide consumers and providers [10]. These possibilities are, however, dependent on future primary studies and meta-analytic research continuing to establish under what circumstances these approaches are most effective, acceptable, and safe.

## Supporting information

**S1 Table. Cochrane risk of bias assessment at the meta-analysis level.**
(DOCX)

**S1 Fig. PRISMA flow diagram.**
(DOCX)

**S2 Fig. Cochrane risk of bias at the study level.** Green = low risk, yellow = unclear risk, red = high risk. Blind = blinding.
(DOCX)

**S3 Fig. PRISMA Checklist.**
(DOCX)

## Acknowledgments

We are grateful to Katty Li and Hannah Reale for their assistance coding primary studies.

## Author Contributions

**Conceptualization:** Simon B. Goldberg, John Torous, Shufang Sun.

**Data curation:** Simon B. Goldberg, Sin U Lam, Otto Simonsson, Shufang Sun.

**Formal analysis:** Simon B. Goldberg.

**Funding acquisition:** Simon B. Goldberg, John Torous, Shufang Sun.

**Investigation:** Simon B. Goldberg, Sin U Lam, Otto Simonsson, John Torous, Shufang Sun.

**Methodology:** Simon B. Goldberg, Sin U Lam, Otto Simonsson, John Torous, Shufang Sun.

**Project administration:** Simon B. Goldberg.

**Software:** Simon B. Goldberg.

**Supervision:** Simon B. Goldberg.

**Visualization:** Simon B. Goldberg.

**Writing – original draft:** Simon B. Goldberg, John Torous.

**Writing – review & editing:** Simon B. Goldberg, Sin U Lam, Otto Simonsson, John Torous, Shufang Sun.

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
