## [Decision Letter · Decision Letter 0]

21 Sep 2021

PDIG-D-21-00022

Mobile phone-based interventions for mental health: A systematic meta-review of 14 meta-analyses of randomized controlled trials

PLOS Digital Health

Dear Dr. Goldberg,

Thank you for submitting your manuscript to PLOS Digital Health. After careful consideration, we feel that it has merit but does not fully meet PLOS Digital Health’s publication criteria as it currently stands. Therefore, we invite you to submit a revised version of the manuscript that addresses the points raised during the review process.

We look forward to receiving your revised manuscript.

Kind regards,

Bridianne O'Dea

Academic Editor

PLOS Digital Health

Leo Anthony Celi

Editor-in-Chief

PLOS Digital Health

Journal Requirements:

1. Please amend detailed Financial Disclosure statement. This is published with the article, therefore should be completed in full sentences and contain the exact wording you wish to be published.

i). State what role the funders took in the study. If the funders had no role in your study, please state: “The funders had no role in study design, data collection and analysis, decision to publish, or preparation of the manuscript.”

2. Please provide separate figure files in .tif or .eps format only, and remove any figures embedded in your manuscript file. If you are using LaTeX, you do not need to remove embedded figures.

For more information about figure files please see our guidelines: https://journals.plos.org/digitalhealth/s/figures

3. We have noticed that you have uploaded supporting information but you have not included a list of legends.  Please add a full list of legends for all supporting information files (including figures, table and data files) after the references list.

4. We notice that your supplementary figures and tables are included in the manuscript file. Please remove them and upload them with the file type 'Supporting Information'. Please ensure that all Supporting Information files are included correctly and that each one has a legend listed in the manuscript after the references list.

Additional Editor Comments (if provided):

We have now received two reviews on your recent submission. We would like to invite you to undertake the major revisions, as outlined by the reviewers. Once this is complete, we will be able to make a decision on your paper.

Reviewers' comments:

Reviewer's Responses to Questions

**Comments to the Author**

1. Does this manuscript meet PLOS Digital Health’s publication criteria? Is the manuscript technically sound, and do the data support the conclusions? The manuscript must describe methodologically and ethically rigorous research with conclusions that are appropriately drawn based on the data presented.

Reviewer #1: Yes

Reviewer #2: Yes

2. Has the statistical analysis been performed appropriately and rigorously?

Reviewer #1: N/A

Reviewer #2: I don't know

3. Have the authors made all data underlying the findings in their manuscript fully available (please refer to the Data Availability Statement at the start of the manuscript PDF file)?

Reviewer #1: Yes

Reviewer #2: Yes

4. Is the manuscript presented in an intelligible fashion and written in standard English?

Reviewer #1: Yes

Reviewer #2: Yes

5. Review Comments to the Author

Reviewer #1: Review comments to the authors

Thank you for the opportunity to review the manuscript titled, Mobile phone-based interventions for mental health: A systematic meta-review of 14 meta-analyses of randomized controlled trials. This meta-review of meta-analyses aimed to determine the strength of evidence showing the effectiveness of mobile phone-based interventions in reducing mental health symptoms. An umbrella review methodology was employed to rate the strength of evidence across each PICO category. They did not find ‘convincing evidence’ regarding efficacy across any of the PICO domains, but did find some ‘highly suggestive evidence’ for small effects across some PICO domains, and conclude mobile-phone based interventions have potential to reduce the societal burden of mental ill-health.

Given the extensive availability of mobile phone-based interventions available to consumers of mental health services, a synthesis of the literature to determine their efficacy is required. This manuscript is well written, uses an appropriate methodology, and draws conclusions consistent with the data presented. I believe significant clarity regarding the terminology used is required. This will improve clarity in regard to the methodology, particularly related to categorisation of conditions for each PICO category, and the data synthesis. Please find specific comments related to each section of the manuscript below.

Introduction: The introduction is well written, reviews the relevant background literature and presents a clear purpose. The following comments are to add clarity.

1. Please define mobile phone-based interventions so it is clear to the reader the types of interventions it includes and use the term consistently. For example, mobile mental health interventions and mobile health interventions also seem to be used but could refer to wearable devices, which are not included in this review.

2. P4, regarding references 18, 19, 20, suggest referring to these as systematic reviews rather than reviews, to distinguish between them and meta-analyses.

3. P5, please provide more detail regarding the Lecomte et al meta-review. Limitations have been identified but it would benefit to reader to understand why the stated limitations might have confounded that study’s conclusions and why the results of the current review might be more valid. To this end, it might help to explain what specifically this review can offer beyond the Lecomte paper.

4. P4-5, I understand the argument that meta-analyses each define PICO differently, which might account for variation in outcomes between meta-analyses as they are essentially asking slightly different research questions. Are the last para on p4 and the second para on p5 saying the same thing?

5. Could more detail be provided regarding the aims of the study? Particularly in regard to what is meant by ‘across PICO categories’. To do this, it may help to provide a very brief description of the method use to achieve this.

Method:

6. Thank you for describing the protocol deviations.

7. Coding of PICO subcategories - comparison conditions. Could the distinction between active, specific and non-specific be clarified? Providing specific examples that include both the intervention and the comparison condition might help (e.g., CBT smartphone app with a comparator of attentional control unlikely to have a therapeutic effect was coded as a non-specific control). I think part of my confusion is that attentional control is used as an example for active and non-specific categories. Could it also be explained why an active category is required, when you have a specific and non-specific active category?

8. Coding of PICO subcategories – intervention. Could the distinction between smartphone app, mobile phoned-based intervention and smartphone interventions be clarified? These terms may need to be revised as ‘mobile phone-based interventions’ seem to be used in the title and intro as an umbrella term capturing all types of mobile phone interventions, but in the method refers to a specific category of interventions.

9. Data synthesis - Could justification of why a representative ES for each unique combination of PICO was required when more than one ES for that unique combination was available? I would have thought that comparisons of these outcomes would be particularly valuable (and rare) in helping to determine homogeneity of results when using the same PICO criteria.

Results:

10. Results of individual studies - Similar to point 9, it is not clear to me why all the available ES are not reported here. I understand the value of presenting a ‘representative’ ES to report on the umbrella methodology category, but it seems valuable to also assess consistency of this outcome if the data is available. I concede, inconsistency may be due to variations in sample size, but potentially other study attributes, such as included study quality, may also contribute.

Discussion: Conclusions drawn are appropriate and limitations included.

11. Were any PICO combinations missing that if evaluated would be valuable contributions to the literature?

12. P17, I understand why ES diminish as the comparison condition becomes more rigorous, but could an explanation be provided regarding why the evidence becomes weaker, i.e., is it specifically because there are too few meta-analyses (or RCTs) employing rigorous comparison conditions?

13. P16 and P17, I believe it would be useful to elaborate on the point that when the comparison condition is a specific active control (rigorous comparison condition) evidence showing non-superiority or a small ES is a not bad outcome, suggesting efficacy is at least equal to other therapeutic interventions.

14. It might be that safety is rarely reported because it is not considered important in mental health interventions. Could its importance be elaborated on in the discussion?

15. A potential limitation is that the interventions were not categorised in terms of the therapeutic model delivered (in addition to intervention quality), e.g., CBT, mindfulness-based CBT and so on. However, I understand given the amount of data being handled and the heterogeneity of RCTs may have made this task difficult.

Reviewer #2: Overall, this is quite a neat paper, well written and thoughtfully constructed. While the initial objective to explore -in this systematic review - states that “mobile phone based interventions have been proposed as a means for reducing the burden of disease associated with mental illness”, this point could be more sufficiently explored/extended in this review. While the discussion section returns to this point, the method and results section do not provide enough evidence to come to this conclusion and -in my opinion - somewhat minimal in statistical evidence. Therefore this conclusion needs to be stated more cautious thereby openly discussing and acknowledging any limitations. The methods and subsequently results section could benefit from the inclusion of some more detailed information on the characteristics of the study characteristics (study outcomes, demographics, intervention type etcetera) of the meta-analysis, and subsequently these variables should be included in the analysis for the current review.

1. It seems like the authors use two terms (EFFICACY and EFFECTIVENESS) to explain their objectives. As per my knowledge, the term EFFECTIVENESS is generally used to describe the results of RCTs in an ideal or controlled environment (laboratory). The term EFFECTIVENESS is more commonly used to describe the effect of an intervention in a natural environment or under real-world conditions. Since the selected RCTs in the meta-analyses included were all conducted in a controlled environment, I am confused by the use of these two terms throughout the manuscript. See also reference to efficacy in the Abstract but throughout the rest of the manuscript effectiveness is discussed. What is the justification for the use of these two terms? To be clarified and/or justified.

2. Although in the first paragraph of the Introduction an effort has been made to explain the growing interest in mobile mental health interventions, this section could benefit from some restructuring, thereby elaborating on research results about the use of smartphone apps across the specific outcome categories, instead of referring to smartphone apps for mental health illness in general (especially since the results are presented as per condition). Adopting a neutral point of view, there is lots of research available examining the use of smartphone applications in the treatment/prevention of specific mental health disorders, that could be discussed here. Also reconsider to reference to smoking; alcohol/drug misuse to explain the burden of disease associated with mental health issues - not always appropriate. Furthermore, although important to emphasize, I am furthermore wondering if the detailed introduction of the use of meta-reviews can be shortened.

3. P4 'While the first generation of these studies focused on feasibility and acceptability, the accumulating evidence

clearly indicates that people suffering from all mental health conditions, including even

schizophrenia, are interested in and willing to use technology as part of their care'

It would be helpful to explain why the reference to schizophrenia was made here.

4. With respect to the eligibility criteria (p7 and elsewhere) and when presenting the results in Table 1/2, it would be helpful to indicate some more detailed information about the interventions examined in the included studies (i.e. interventions of preventative nature; behavioral etcetera) and use type of intervention as a comparison between studies, in addition to the use of the current categorization of interventions (i.e. text messaging, smartphone intervention, meditation apps etcetera) . This should be taken into account when analysis the results and could potentially influence the results found.

5. In the Methods and Results it furthermore doesn't mention how clinical condition was determined for each of the samples in the included studies (definition of clinical diagnosis; subclinical/clinical). It would be helpful if this was clarified further.

6. P7. Databases were searched since inception until October 31st, 2020.

Would be worth to update search to a more recent date.

7. P7/P8 - In the initial search for relevant meta-analysis the search terms digital health and e-health weren't included? Consider updating search terms when exploring recent publications.

8. The authors should comment on whether or not the raters conducted screening/rating/data extraction independently AND blindly (p8). It would be helpful if this was clarified further.

9. To be able to summarize the includes studies based upon the PICO principles population and condition was coded. sample population was coded. It would be helpful to clarify coding adults vs adolescents and how the clinical condition coding was established -this is important to know in the interpretation of the results. I would furthermore suggest to include more information about the age distribution across studies and include this in the Results section and Tables.

10. Summary Measures P9. It is stated that "Standardized mean difference (i.e., Cohen’s d, Hedges’ g) served as our primary effect' - So both Cohen's and hedges are reported? Or all effect sizes converted to either hedges/cohen? I am assuming the latter happened for subsequent analysis but should be clarified. For example, Figure 2 states cohen but unclear for ES in Table 2. It is furthermore unclear when effect sizes to be considered as not-eligible effect sizes to be included in this review.

11. Synthesis the Results P10. I could not find any information about time points outcomes assessed? What about variations in follow-up period? Are effect sizes corrected for that? Would be helpful to discuss this in the Methods and Results.

12. P11 - Risk of bias Within Studies. It would be good to have the results of Supplemental Materials Table 1 explained a little more here - consider Eggers regression test or presenting p values.

13. In line with my previous comments, Table 1 includes a limited amount of information about each meta-analysis. I think more information about demographics where possible (such as age range, gender), clinical conditions, what instruments used to measures outcomes, as well as characteristics of the interventions, should be included in order to permit a more in depth critical appraisal of the studies and inform discussions with respect to the representativeness of the findings and draw any conclusions about effectiveness of the smartphone 'intervention' examined.

14. Please furthermore ensure the index of Table 2 explains all abbreviations included in the Table.

15. A PRISMA checklist should be included as supplementary material or as an appendix.

6. PLOS authors have the option to publish the peer review history of their article (what does this mean?). If published, this will include your full peer review and any attached files.

**Do you want your identity to be public for this peer review?** For information about this choice, including consent withdrawal, please see our Privacy Policy.

Reviewer #1: No

Reviewer #2: No

---

## [Editor Report · Decision Letter 1]

27 Oct 2021

Mobile phone-based interventions for mental health: A systematic meta-review of 14 meta-analyses of randomized controlled trials

PDIG-D-21-00022R1

Dear Dr. Goldberg,

We're pleased to inform you that your manuscript has been judged scientifically suitable for publication and will be formally accepted for publication once it meets all outstanding technical requirements.

Within one week, you'll receive an e-mail detailing the required amendments. When these have been addressed, you'll receive a formal acceptance letter and your manuscript will be scheduled for publication.

An invoice for payment will follow shortly after the formal acceptance. To ensure an efficient process, please log into Editorial Manager at https://www.editorialmanager.com/pdig/ click the 'Update My Information' link at the top of the page, and double check that your user information is up-to-date. If you have any billing related questions, please contact our Author Billing department directly at authorbilling@plos.org.

Kind regards,

Bridianne O'Dea

Academic Editor

PLOS Digital Health

Additional Editor Comments (optional):

Thank you for your patience in the re-review of your revised manuscript. Your detailed response is appreciated. I am satisfied with your responses alongside the changes to the manuscript. I am pleased to accept your review.